# Physicians and Machine-Learning Algorithm Performance in Predicting Left-Ventricular Systolic Dysfunction from a Standard 12-Lead-Electrocardiogram

**DOI:** 10.3390/jcm11226767

**Published:** 2022-11-15

**Authors:** Tomer Golany, Kira Radinsky, Natalia Kofman, Ilya Litovchik, Revital Young, Antoinette Monayer, Itamar Love, Faina Tziporin, Ido Minha, Yakir Yehuda, Tomer Ziv-Baran, Shmuel Fuchs, Sa’ar Minha

**Affiliations:** 1Taub Faculty of Computer Sciences, Technion—Israel Institute of Technology, Haifa 3200003, Israel; 2Sackler School of Medicine, Tel-Aviv University, Ramat-Aviv, Tel Aviv 6997801, Israel; 3Department of Cardiology, Shamir Medical Center, Be’er-Yaakov 7033001, Israel; 4Department of Mathematics and Computer Science, The Open University, Raanana 4353701, Israel; 5Department of Epidemiology and Preventative Medicine, School of Public Health, Sackler Faculty of Medicine, Tel Aviv University, Tel Aviv 6997801, Israel

**Keywords:** heart failure, machine learning, artificial intelligence, electrocardiogram, early diagnosis

## Abstract

Early detection of left ventricular systolic dysfunction (LVSD) may prompt early care and improve outcomes for asymptomatic patients. Standard 12-lead ECG may be used to predict LVSD. We aimed to compare the performance of Machine Learning Algorithms (MLA) and physicians in predicting LVSD from a standard 12-lead ECG. By utilizing a dataset of 13,820 pairs of ECGs and echocardiography, a deep residual convolutional neural network was trained for predicting LVSD (ejection fraction (EF) < 50%) from ECG. The ECGs of the test set (*n* = 850) were assessed for LVSD by the MLA and six physicians. The performance was compared using sensitivity, specificity, and C-statistics. The interobserver agreement between the physicians for the prediction of LVSD was moderate (κ = 0.50), with average sensitivity and specificity of 70%. The C-statistic of the MLA was 0.85. Repeating this analysis with LVSD defined as EF < 35% resulted in an improvement in physicians’ average sensitivity to 84% but their specificity decreased to 57%. The MLA C-statistic was 0.88 with this threshold. We conclude that although MLA outperformed physicians in predicting LVSD from standard ECG, prior to robust implementation of MLA in ECG machines, physicians should be encouraged to use this approach as a simple and readily available aid for LVSD screening.

## 1. Introduction

Heart failure (HF) is a major global health problem, associated with high morbidity and mortality. It is estimated that 6.2 million Americans >20 years of age suffer from some form of HF [1]. Asymptomatic left-ventricular systolic dysfunction (LVSD) is reported to be more common than symptomatic HF and is associated with poor outcomes [2,3,4,5]. Several interventions have been established as effective in terms of improving prognoses once asymptomatic LVSD has been diagnosed [6,7]. Thus, early diagnosis of asymptomatic LVSD is of significant clinical and prognostic value. Echocardiography, a non-invasive, ultrasound-based exam, is the recommended key investigation for the assessment of cardiac function [8]. Unfortunately, this exam is less accessible, especially for primary care physicians, and requires professional interpretation skills, making it less relevant as a screening test [9,10,11]. A screening strategy based on B-type natriuretic peptide, a serum protein which is elevated in patients with overt HF, has established a role in asymptomatic LVSD screening, although the performance of this metric is not robust [12,13,14,15,16]. As opposed to the two aforementioned tests, electrocardiography (ECG) is a readily available, non-invasive, basic cardiovascular test which is performed and interpreted over 300 million times annually. This makes ECG an attractive test for asymptomatic LVSD screening. The use of machine learning algorithms (MLA), a subset of artificial intelligence, in healthcare is constantly increasing due to their ability to predict specific outcomes from various types of data. Such algorithms were recently reported to have high performance in predicting low ejection-fraction (≤35%) in echocardiography from a standard 12-lead electrocardiogram (ECG) [17,18]. Assuming ECG may be utilized for the prediction of LVSD by MLA, it is interesting to explore if physicians can predict LVSD by analyzing a standard ECG. We hypothesize that both MLA and physicians can predict abnormal EF from a standard 12-lead ECG.

## 2. Materials and Methods

### 2.1. Study Population

To test the hypotheses, a paired ECG and echocardiography dataset was created. After obtaining the approval of the local institutional review board at Shamir Medical Center, the hospital’s databases were used to identify all patients that had a standard 10-s, 12-lead ECG recorded with an FDA-approved ECG machine (CardiMax FX-8222/FX-8322/FX8200 Fukuda Denshi Co. Ltd., Tokyo, Japan) and had undergone a standardized echocardiography in our center. These data were anonymized. Due to this and the retrospective nature of the dataset, patient consent was waived. The billing code for adult echocardiography was used to exclude patients under the age of 18 years. The initial dataset was censored to include pairs of tests with a time interval of ≤180 days. This cohort was split into three datasets as follows: the “Test dataset” included 1000 randomly collected patients of which 500 had normal EF, the “training dataset” included 80% of the dataset and the “validation dataset” included the remaining pairs. The test set included an equal number of normal and abnormal EF cases (and not a random collection of cases that may not have included a sufficient number of abnormal EF cases) to establish the true discriminative performance of both physicians and the MLA. After data collection, to increase the clinical relevancy of the study, we further censored all test pairs performed within more than 14 days from each other. In the case of multiple pairs within this time frame, the pair with the shortest time interval was considered valid for analysis. The training dataset was used for the learning process of MLA, the validation dataset was used to tune the hyperparameters and the test dataset was used to evaluate the performance of both the physicians and MLA in predicting EF based on ECG. Echocardiography data included standard semi-quantitative assessments of the left-ventricular EF. EF was labeled as either “normal” for EF ≥ 50% or “abnormal” if EF < 50%. This cut-off was based on the European Society of Cardiology definition for preserved left-ventricular EF [8]. Demographic and baseline clinical profiles of the included patients were obtained from the electronic medical record, based on the 9th International Classification of Diseases.

### 2.2. Physician Predictions of EF by ECG

The ECGs in the test dataset were evaluated by six physicians after data anonymization. The six physicians included two senior cardiologists with over 20 years of practice, one senior cardiologist with 12 years of practice, and three cardiology fellows with 2–7 years of experience as physicians. The physicians were asked to state whether they believed that the EF of the patient with each ECG was normal or abnormal and were not allowed to consult with other physicians.

### 2.3. Deep Residual Convolutional Neural Network for EF Prediction

A deep Residual Convolutional neural network (ResNet) was utilized for the prediction of EF from ECG. Inspired by He et al. [19], ResNet is a form of MLA, consisting of multiple hidden layers comprising a series of convolutional layers. All ECG records were resampled to 500 Hz. After resampling and padding, all ECG records were represented by a tensor matrix of shape 12 × 5000 mV, where the first dimension represents the spatial dimension, and the second dimension represents the temporal dimension. That is, each lead of a 12-lead ECG is considered a 10 s signal sampled at 500 Hz. ECG samples that were below 10 s, were zero-padded [20], and ECG signals longer than 10 s were sliced. This tensor matrix was the input of the ResNet. The training set was used to train the model in classifying ECG as either normal or abnormal EF. The ResNet output is the probability of the ECG to be normal/abnormal.

The network architecture which empirically provided the best results (validated using 10-fold cross-validation using the training data) consisted of a convolution layer, followed by a max-pooling layer, followed by six residual blocks (Appendix A). Each residual block consisted of three convolution layers, and between each convolution layer, batch-normalization [21] and Rectified Linear Unit (Relu) [22] activation were performed. A skip connection [19] was applied between the input of the block to the output of the third convolution layer. The output of the last residual block was fed into a global average pooling layer, followed by a dense layer with a SoftMax [23] activation function to predict output class probabilities. The first convolution layer had 16 filters of size 7 × 7. The residual blocks started with 16 filters, increasing to 32 filters in the last block. The size of the kernel in the residual blocks started at 5 × 5 and decreased to 3 × 3. In all the residual blocks except the first one, the first convolution layer down-sampled the input temporal dimension by a stride of 2.

The network was trained by feeding 12-lead ECG batches of size 16 from the training set. The categorical cross-entropy [24] loss was minimized using an Adam optimizer [25] with an initial learning rate of 0.0001. The training set ran for 100 epochs, with the final model being the one with the best accuracy on the validation set. Several architectures and hyperparameters were explored by utilizing an iterative process, and the hyperparameters yielded the best results on the validation set. We tuned the hyper-parameters as follows: batch size (0.01, 0.001, 0.0001), optimizer algorithm (RMS-Prop [26], Adam [25], Stochastic Gradient Descent [27]), and number of residual blocks [3,4,5,6,7].

We also attempted the following alternative architectures: (1) Temporal convolution network with increasing dilation rate, inspired by Oord [28]. (2) Long Short-Term Memory blocks above convolutional layers [29]. (3) Deep neural network with temporal 1d-kernels to learn the features within each lead, followed by spatial 1d-kernels which fused the data from all leads. These methods did not yield statistically significant improvement compared to the ResNet we used in this work.

### 2.4. Statistical Analysis

Continuous data are presented as mean and standard deviation while median and interquartile range (IQR) are used if violating normal distribution. The Kolmogorov–Smirnov test was used to assess the normality of distribution. Categorical data are presented as numbers and percentages. An independent sample two-sided *t*-test or Mann-Whitney test was used to compare continuous variables, and the χ^2^ was used to compare categorical values. The diagnostic performance of the physicians in predicting LVSD from ECG is described by sensitivity and specificity. Fleiss’ kappa coefficient with a 95% confidence interval was calculated to evaluate the interobserver agreement. The LVSD prediction of the MLA is presented by a receiver operator curve and C-statistic (area under the ROC). *p* value < 0.05 was considered significant. Statistical analyses utilized SPSS software (IBM SPSS statistics for windows, version 27.0.1.0, IBM Corp., Armonk, NY, USA, 2020).

## 3. Results

### 3.1. Baseline Characteristics

The initial dataset of paired ECG and echocardiographic exams consisted of 200,879 pairs (Figure 1). After excluding pairs of exams performed outside of a 14-day period and those that had more than one ECG or one echocardiographic exam within this timeframe (i.e., only one pair per patient with the shortest time gap between tests was included), the final dataset included 13,820 pairs with a median time interval of one day (IQR 0–3) between the ECG and the echocardiography test. The median age and EF of the included patients were 69.9 years (59.7–80.2), and 55% (40–60) respectively; 61% of patients were male. Past medical history data were available for 12,657 (91.6%) patients. Diabetes mellitus and hypertension were present in 54.8% and 76.1% of cases, respectively. While no significant differences were noted in the baseline characteristics between the validation and training datasets, the test dataset, in which half of the patients had abnormal LV function, had a higher prevalence of male gender, peripheral and coronary artery disease and lower median EF (Table 1).

As demonstrated in Table 2, compared with patients with normal EF, patients with LVSD were more likely to be males with a higher incidence of cardiovascular risk factors including, among others, diabetes mellitus, hypertension, and obesity.

### 3.2. LVSD Prediction

The performance of the MLA model and the physicians for predicting LVSD is presented in Figure 2 and Table 3.

The C-statistic of the MLA was 0.85. Significant variability was noted in the performance of physicians with moderate interobserver agreement (Fleiss’ kappa = 0.50; 95% CI 0.48–0.52). We further compared the interobserver agreement between the two physicians with the best performance (#6 and #2), yielding a kappa of 0.58 (95% CI 0.53–0.63). Figure 3 displays four examples of discordance between all six physicians and the MLA model for the prediction of LVSD.

To simplify comparisons between the physicians and the MLA model, the average performance of the physician was calculated to have a sensitivity and specificity of 70%. This was found to be lower than the MLA (sensitivity and specificity of 78% by den index). When this was repeated for the three senior cardiologists, the sensitivity improved to 77.6% but the specificity decreased to 64.7%.

To further compare the performance of MLA and the physicians, all the cases in the test set were re-labeled, defining EF ≤ 35% as LVSD, while the rest were coded as normal. A comparison of the baseline characteristics between patients with EF ≤ 35% and those with EF > 35% is presented in Appendix A. The C-statistic for the MLA was 0.88 for predicting EF ≤ 35% (Figure 4). The performance of the average physician improved to 84% sensitivity, but the specificity decreased to 57%. When the average performance of the senior cardiologists was calculated, a sensitivity of 90.7% was recorded alongside a decreased specificity of 51.9%. The performance of the MLA did not change significantly (78% sensitivity and 79% specificity by utilizing the Youden index).

## 4. Discussion

The main results of this study indicate that: (1) Physicians can predict LVSD from a standard 12-lead ECG with moderate interobserver variability; (2) MLA outperformed physicians in predictions of LVSD; and (3) While the sensitivity of physicians in terms of predicting LVSD improved significantly when the threshold was set to EF ≤ 35%, the performance of MLA was consistent across different LVSD thresholds.

Despite major advances in pharmacological and interventional care, heart failure is still associated with dire outcomes and significant social and financial burdens. It is thus of paramount importance to implement novel screening methods aimed at early detection. Early diagnosis of patients at the asymptomatic phase can facilitate care and follow-up which, in turn, may be associated with improved outcomes. Standard 12-lead ECG is a non-invasive, readily available, affordable test performed on millions of people each year which may be of value as a screening aid for depressed EF. A prior attempt to associate ECG and HF was reported by analyzing data from the Multi-Ethnic Study of Atherosclerosis (MESA) trial [30]. In that report, and other studies, specific ECG features (e.g., axis deviation, QRS duration, ST-T segment abnormalities, and bundle-branch blocks) were independently associated with HF diagnosis [31,32,33]. More recently, “advanced ECG” (aECG) was demonstrated to have high specificity and sensitivity for LVSD diagnosis. This technology utilizes computerized analyses of the standard 12-lead ECG that incorporate spatial and temporal data that exceed the data provided by the standard ECG analysis software [34,35]. Of note, a relatively small-scale study demonstrated that ECG outperformed physicians in predicting LVSD in a cohort of 79 patients [36]. Thus, although significant advances have been made in the field of ECG and its utilization for LVSD diagnosis, at present, echocardiography is considered the standard test for HF screening.

In recent years, the introduction of artificial intelligence (AI) to medical research has allowed the implementation of algorithms aimed at improving diagnostic ability and decision making. Examples include the implementation of AI algorithms for fully automated interpretation of both echocardiography and cardiac magnetic resonance [37,38]. Several studies have focused on the ability of AI to interpret ECG abnormalities from both single and 12-lead ECG [20,39]. Other studies have demonstrated the superior performance of AI compared to physicians regarding ECG interpretation [40,41,42].

This was also demonstrated for predicting LVSD from ECG. In a study by Attia et al., an AI algorithm was used in a cohort of 97,892 paired ECG and echocardiograms to discriminate EF ≤ 35% or >35% [17]. A C-statistic of 0.93 was achieved in this study, and these results were later validated on a separate cohort of patients [43]. Similar results were obtained from Kwon et. al, who demonstrated a C-statistic of 0.84 for the prediction of EF ≤ 40% and 0.82 for EF ≤ 50% from a standard ECG [18]. A later report from this group demonstrated similar performance for EF ≤ 40% prediction from both 12-lead and single-lead ECG [44]. The performance of MLA in the present analysis for predicting EF < 50% from a standard 12-lead ECG is in line with these reports, and it can be speculated that the performance variability between the reports depends on the sample size and the threshold used to label EF as abnormal, among other indices [45]. The implementation of the AI algorithm for EF prediction was recently tested in the “ECG AI-Guided Screening for Low Ejection Fraction” (EAGLE) randomized controlled trial [46]. This study demonstrated that the use of AI-guided ECG-based prediction of EF in clinical practice was associated with a high incidence of referral to echocardiography compared with standard of care.

Since at present, MLA is not routinely embedded into ECG interpretation, and most ECGs are being analyzed and interpreted by physicians, it is of interest to explore the question of whether physicians can predict EF from ECG in a similar manner as MLA. Similarly to Hannun et al. [40], we calculated the average sensitivity and specificity of physicians for predicting LVSD to be 70% for both. The MLA outperformed the average physician with 78.3% sensitivity and 78.1% specificity by utilizing the Youden index to determine the best performance. Interestingly, when the threshold for EF abnormality was set to ≤35%, physician sensitivity improved to 84% with 57% specificity, but the performance of the MLA did not change significantly (78% sensitivity; 79% specificity). This was even more pronounced in the performance of the senior cardiologists, in which the sensitivity increased from 77.6% to 90.7%, practically outperforming the MLA. The significant improvement in the sensitivity of the physicians when using a lower threshold for defining LVSD may indicate that the ECG features used by physicians are more relevant to rule out low ejection fraction, while their ability to identify potentially low EF is less robust. On the other hand, although outperformed by MLA, physicians’ performance for both retaining and excluding LVSD when a threshold of EF < 50% was used was as balanced as that of MLA. As we demonstrated, the interobserver agreement between physicians for EF prediction was moderate at best (Fleiss kappa = 0.50; 95% CI 0.48–0.52), even when two physicians with the best performance were studied (Fleiss kappa = 0.58; 95% CI 0.55–0.61). This may indicate that the labeling of EF as abnormal by physicians is based on the different weights given to the ECG segment. For example, while one physician may place significant weight on the existence of Q waves when predicting an EF as abnormal, the other might prioritize the ST-T segment over Q waves.

Taken together, since echocardiography is still the recommended key investigation exam for the diagnosis of cardiac dysfunction, and while this test is not readily available, before the broad implementation of streamlined MLA-based LVSD prediction, physicians should be encouraged to screen for LVSD whenever an ECG is interpreted.

Several limitations to this study should be acknowledged. First, the parameters used by MLA for the classification of ECGs in this study are unknown. This inherent “black box” feature of MLA prevented us from highlighting specific traditional (e.g., bundle branch blocks, Q waves, etc.) and non-traditional, unknown parameters associated with LVSD. Beyond the inability to gain new insights regarding unknown features associated with LSVD, this lack of identifiable parameters further limited our ability to retrain physicians and improve their performance. Although less likely in a large cohort, it is also possible that some of the determinants for classification by the MLA are based on artifacts or features which are relevant only to the machines used in this study. Limitations of this type concerning the use of MLA have been discussed and highlighted by prior reports in this field of research [47,48,49]. Similarly, although physicians demonstrated fair performance for LVSD prediction, since only six physicians participated in the study, we cannot point out which ECG features were the most influential for labeling. These important questions are to be explored in a separate study.

Second, physician performance was based on six physicians, three of whom were senior cardiologists while the others were cardiology fellows. Due to this selection bias, perhaps different performance levels would have been demonstrated if a larger number of participating physicians from different disciplines with various years of experience had been included. On the other hand, it seems reasonable that cardiologists and cardiology fellows represent the population of physicians with the most experience and exposure to both ECG and echocardiography, and thus, it is believed that this cohort represents the upper tier of performance by physicians. Lastly, the MLA models for predicting LVSD from ECG were based on a large cohort of patients that had undergone both tests at a large medical center. This exposes our results to selection bias. This is also relevant for the use and interpretation of MLA based analyses in general, since these models are based on the data presented to the model for learning, which may not necessarily represent the general population. Although we believe in the robustness of the presented data, an external validation study is needed before broadly implementing this model for screening for LVSD.

## 5. Conclusions

Although outperformed by MLA, physicians can predict LVSD from a standard 12-lead ECG. This finding should promote the use of ECG as a screening aid for the early detection of asymptomatic LVSD.

## Figures and Tables

**Figure 1 jcm-11-06767-f001:**
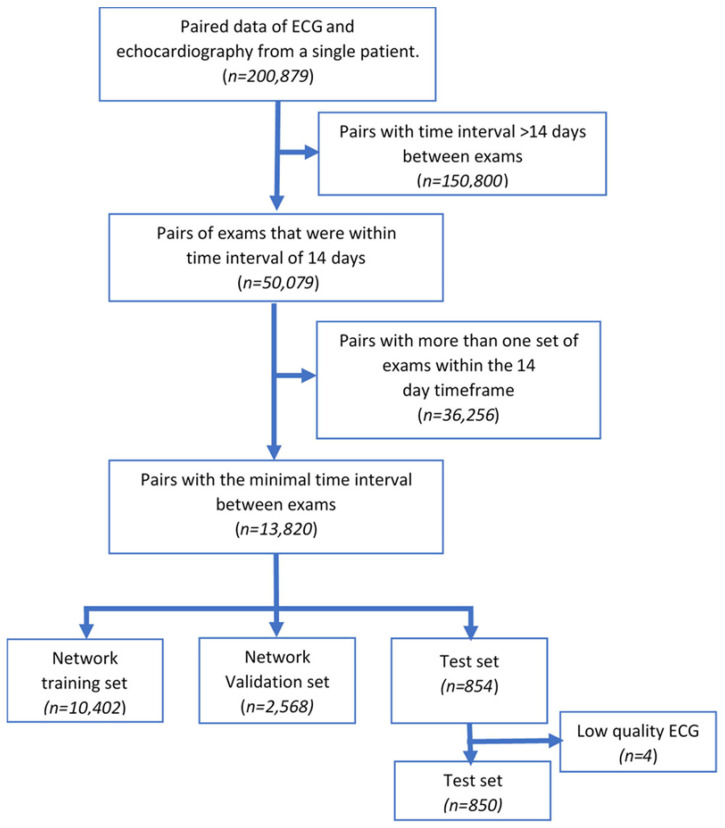
Study flow diagram.

**Figure 2 jcm-11-06767-f002:**
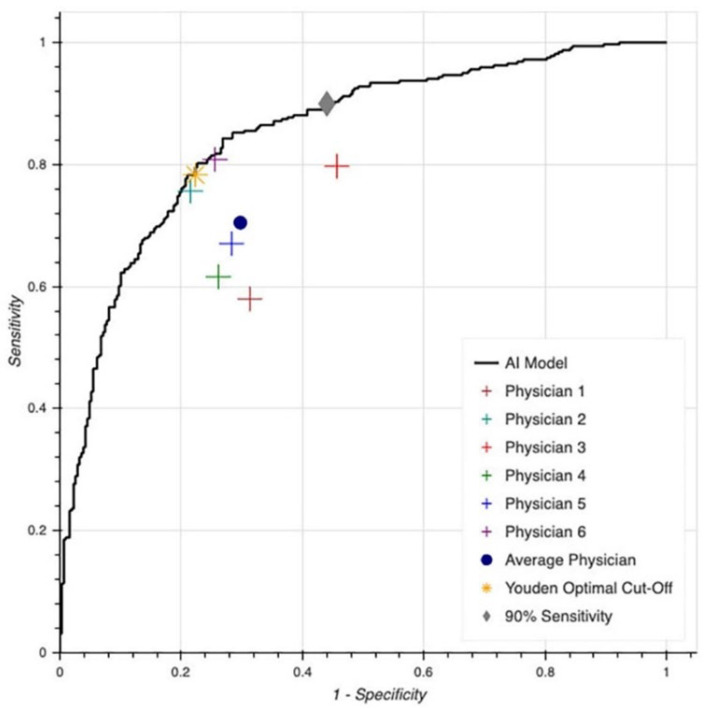
Receiver operating characteristic curve and physician performance for predicting left ventricular systolic dysfunction from 12-lead ECG (Area under the curve = 0.85).

**Figure 3 jcm-11-06767-f003:**
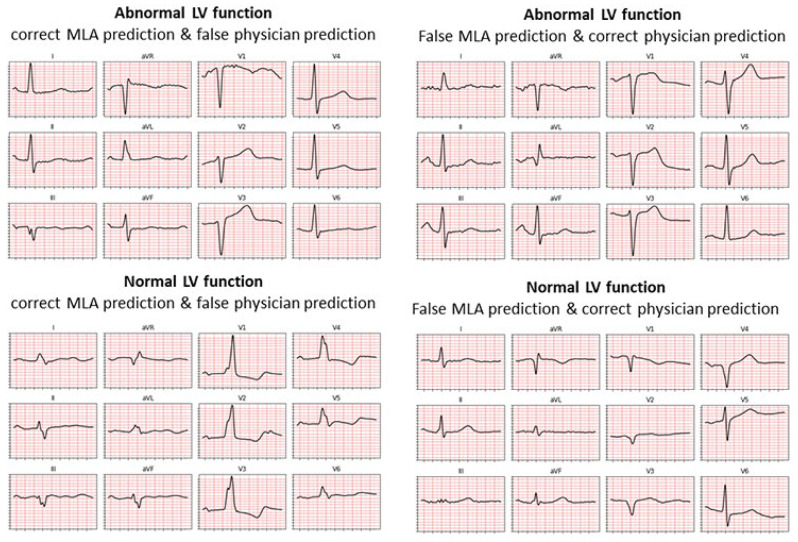
ECG examples of discordant diagnosis of normal/abnormal left ventricular systolic dysfunction by physicians and machine learning algorithm.

**Figure 4 jcm-11-06767-f004:**
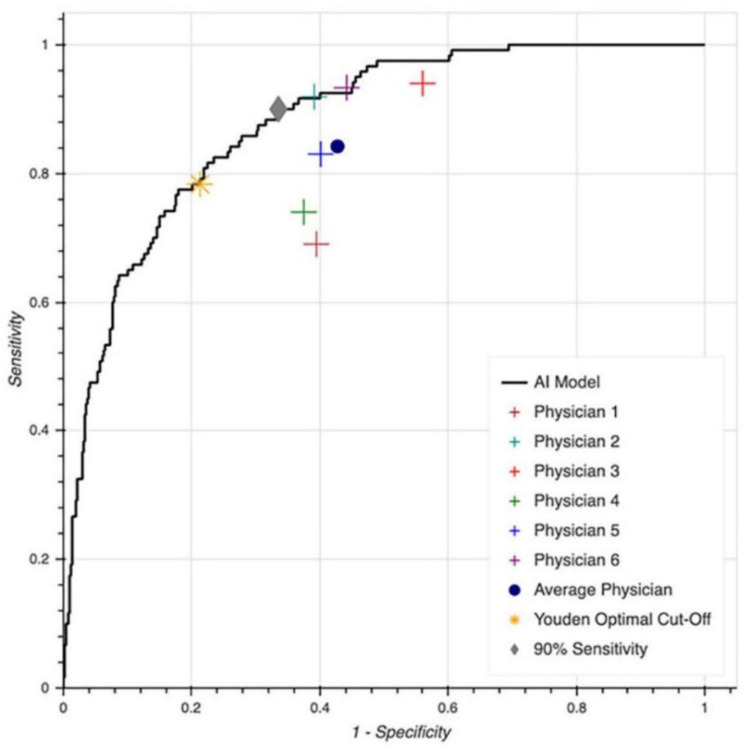
Receiver operating characteristic curve and physician performance in differentiating normal from abnormal ejection-fraction form 12-lead ECG by defining EF ≤ 35% as abnormal (area under curve = 0.88).

**Table 1 jcm-11-06767-t001:** Baseline characteristics of three datasets of echocardiography and ECG pairs.

Index	Training Dataset (*n* = 10,402)	Validation Dataset (*n* = 2568)	Test Dataset (*n* = 850)	*p*-Value
Age (years [IQR])	69.8 [59.6–80.1]	70.0 [59.4–80.2]	71.3 [61.1–81.4]	0.07
Male (*n*; %)	6347; 61.0%	1569; 61.1%	560; 65.9%	0.02
Family history of CAD	2591; 27.1%	590; 25.1%	224; 29.2%	0.04
Diabetes mellitus (*n*; %)	5254; 54.9%	1272; 54.2%	423; 54.8%	0.80
Hypertension (*n*; %)	7236; 75.8%	1804; 76.9%	587; 76.6%	0.50
Hyperlipidemia (*n*; %)	6007; 63.9%	1509; 64.3%	511; 66.7%	0.68
Chronic kidney disease (*n*; %)	3426; 35.9%	806; 34.4%	310; 40.5%	0.09
Chronic dialysis	2490; 26.1%	571; 24.3%	215; 28.1%	0.08
Peripheral vascular disease	2780; 29.1%	658; 28.0%	256; 33.4%	0.02
Obesity (*n*; %)	2569; 37.4%	856; 36.9%	283; 36.9%	0.88
Atrial fibrillation/flutter	3653; 38.3%	839; 35.8%	314; 41.0%	0.02
Pacemaker	2607; 27.3%	592; 25.2%	233; 30.4%	0.01
Smoking history	4351; 45.6%	1052; 44.8%	367; 47.9%	0.33
COPD	2977; 31.2%	716; 30.5%	263; 34.3%	0.13
Stroke/TIA	2983; 31.3%	673; 28,7%	244; 31.9%	0.04
Ischemic heart disease	4479; 46.9%	1084; 46.2%	408; 53.3%	<0.001
Coronary artery bypass surgery	2729; 28.6%	632; 26.9%	247; 32.2%	0.02
PCI	2880; 30.2%	675; 28.8%	251;32.8%	0.10
EF (%) median [IQR]	55 [40–60]	55 [40–60]	50 [40–60]	<0.001
EF ≥ 50% (*n*; %)	6729; 64.7%	1642; 63.9%	427; 50.2%	<0.001

Legend: IQR—Interquartile range; CAD—Coronary artery disease; COPD—Chronic obstructive pulmonary disease; TIA—Transient ischemic accident; PCI—Percutaneous coronary intervention; EF—Ejection fraction.

**Table 2 jcm-11-06767-t002:** Baseline characteristics of patients with normal ejection fraction (EF ≥ 50%) and abnormal ejection fraction.

Index	EF ≥ 50% (*n* = 8798)	EF < 50% (*n* = 5022)	*p*-Value
Age (median years [IQR])	70.1 [58.8–80.6]	69.9 [60.9–79.7]	0.26
Male (*n*; %)	4797; 54.5%	3679; 73.3%	<0.001
Family history of coronary artery disease (*n*; %)	1640; 20.9%	1765; 36.8%	<0.001
Diabetes Mellitus (*n*; %)	3855; 49.0%	3082; 64.3%	<0.001
Hypertension (*n*; %)	5833; 74.2%	3794; 79.2%	<0.001
Chronic kidney disease (*n*; %)	2301; 29.3%	2241; 46.8%	<0.001
Dialysis (*n*; %)	1558; 19.8%	1720; 35.9%	<0.001
Peripheral vascular disease (*n*; %)	1810; 23.0%	1885; 39.3%	<0.001
Hyperlipidemia (*n*; %)	4698; 59.7%	3329; 69.5%	<0.001
Obesity (*n*; %)	2568; 32.9%	2131; 44.5%	<0.001
Atrial fibrillation/flutter (*n*; %)	2591; 32.9%	2215; 46.2%	<0.001
Pacemaker (*n*; %)	1639; 20.8%	1793; 37.4%	<0.001
Chronic obstructive pulmonary disease (*n*; %)	2034; 25.9%	1922; 40.1%	<0.001
Smoking history	2984; 37.9%	2786; 58.1%	<0.001
Stroke/Transient ischemia accident (*n*; %)	1971; 25.1%	1929; 40.2%	<0.001
Ischemic heart disease	2924; 37.2%	3047; 63.6%	<0.001
Coronary artery bypass surgery (*n*; %)	1683; 21.4%	1925; 40.2%	<0.001
Percutaneous coronary intervention (*n*; %)	1824; 23.2%	1982; 41.4%	<0.001
Ejection fraction (median % [IQR])	60 [55–60]	40 [30–40]	<0.001

EF—Ejection fraction; IQR—Interquartile range.

**Table 3 jcm-11-06767-t003:** Physicians and Machine learning algorithm for predicting abnormal ejection fraction (%) from a standard 12-lead ECG.

	Predicting EF < 50%		Predicting EF ≤ 35%	
	Sensitivity	Specificity	Sensitivity	Specificity
Physician 1	59.4%	67.1%	69.6%	59.5%
Physician 2	76.1%	76.1%	90.0%	60.0%
Physician 3	80.8%	53.3%	93.2%	43.4%
Physician 4	63.2%	73.9%	75.8%	62.9%
Physician 5	69.2%	69.7%	83.9%	58.5%
Physician 6	82.7%	71.1%	95.0%	53.8%
Average physician	71.9%	68.5%	84.6%	56.3%
Average senior physician	77.6%	64.7%	90.7%	51.9%
MLA performance based on Youden cut point	78.3%	78.1%	78%	79%
MLA performance based on 90% sensitivity	90%	55.7%	90%	66%

EF—Ejection fraction; MLA—Machine learning algorithms.

## Data Availability

The data presented in this study are available on request from the corresponding author. The data are not publicly available due to privacy.

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
