# Peer review of "Physicians and Machine-Learning Algorithm Performance in Predicting Left-Ventricular Systolic Dysfunction from a Standard 12-Lead-Electrocardiogram"

_jcm, 2022, doi:10.3390/jcm11226767_

Round 1

Reviewer 1 Report

In the present paper (Physicians and machine-learning algorithm performance in predicting left-ventricular systolic dysfunction from a standard 12-lead-electrocardiogram), authors expose the usefulness of a Machine-learning algorythm (MLA) in predicting Left Ventricle dysfunction (LVD), compared to a group of cardiologists. This is such an interesting work, with quite some interesting insights in how ML can improve the day a day clinical practice.

Authors find that MLA can outperform cardiologist in detecting patients with LVD based only in a 12-lead ECG. But, although the present work expose a new and interesting idea and in general lines it is well carried, there are still some issues that if addressed can certainly improve this work.

Major concerns:

1) The database used in this work is based on patients that underwent an echocardiography. It can lead to a selection bias, since the previous probability of LVD is really high (moreover, in table 1 can be seen a high prevalence of cardiovascular risk factors). Thus, it can make the MLA performance to be higher than in real clinical practice. This should be included as a limitation of the study.

2) Physicians included in the work have a high range of clinical experience, and a significant part of them are still in training (there are only 3 of 6 senior cardiologists). This can make the clinician performance in detecting LVD to be lower than it really is. Moreover, authors describe a significant inter-observer variability, which improves when taking into consideration clinicians #6 and #2 (who are the ones with better performance, based on the ROC curves). In this sense, author should present a separated analysis of clinical performance based only in senior cardiologists, with significant clinical experience.

Minor concerns

1) In lines 40-41, authors address that echocardiography (ECO) is not readily accessible in clinical practice. But this is not completely true, since most recent recommendations encourage to make ECO available in every cardiovascular clinic to complete the routine cardiovascular examination. Authors should eliminate this statement.

Author Response

Major concerns:

1) The database used in this work is based on patients that underwent echocardiography. It can lead to a selection bias, since the previous probability of LVD is really high (moreover, in table 1 can be seen a high prevalence of cardiovascular risk factors). Thus, it can make the MLA performance to be higher than in real clinical practice. This should be included as a limitation of the study.

Reply: This is an important limitation. We’ve modified the limitations section and the relevant paragraph now read:
Lastly, the MLA models for predicting LVSD from ECG were based on a large cohort of patients that underwent both tests at a large medical center. This exposes our results to a selection bias. Although we believe in the robustness of the presented data, an external validation study is needed before a broad implementation of this model for screening for LVSD.

2) Physicians included in the work have a high range of clinical experience, and a significant part of them are still in training (there are only 3 of 6 senior cardiologists). This can make the clinician performance in detecting LVD to be lower than it really is. Moreover, authors describe a significant inter-observer variability, which improves when taking into consideration clinicians #6 and #2 (who are the ones with better performance, based on the ROC curves). In this sense, author should present a separated analysis of clinical performance based only in senior cardiologists, with significant clinical experience.

Reply: As per the reviewer’s request, we’ve calculated and added the performance of the three senior physicians. This is presented in Table 3 and the text as follows: “To simplify the comparison between the physicians and the MLA model, the average performance of the physician was calculated to have a sensitivity and specificity of 70%. This was found to be lower than the MLA (sensitivity and specificity of 78% by den index). When this was repeated for the three senior cardiologists, the sensitivity improved to 77.6% and but the specificity decreased to 64.7%. To further compare the performance of MLA and the physicians, all the cases in the test set were re-labeled defining EF≤35% as LVSD while the rest were coded as normal. The comparison of the baseline characteristics between patients with EF≤35% and those with EF>35% is presented in Table S1. The C-statistic for the MLA was 0.88 for predicting EF≤35% (figure 4). The performance of the average physician improved to 84% sensitivity, but the specificity decreased to 57%. When the average performance of the senior cardiologists was calculated, a sensitivity of 90.7% was recorded alongside with decreased specificity of 51.9%. The performance of the MLA did not change significantly (78% sensitivity and 79% specificity by utilizing the Youden index).“ Further, these results were discussed in the discussion section (line 279) as follows: “This was even more pronounced in the performance of the average senior cardiologists in which the sensitivity increased from 77.6% to 90.7%, practically outperforming the MLA

Minor concerns

1) In lines 40-41, authors address that echocardiography (ECO) is not readily accessible in clinical practice. But this is not completely true, since most recent recommendations encourage to make ECO available in every cardiovascular clinic to complete the routine cardiovascular examination. Authors should eliminate this statement.

Reply: It is completely true that echocardiography is a recommended and encouraged test but unfortunately, this test is less accessible than ECG, especially for primary care physicians. We opted to modify this phrase and support it with relevant citations as follows:
“Unfortunately, this exam is not less accessible, especially for primary care physicians. and requires professional interpretation skills and thus, is less relevant as a screening test.[9-11]”

Reviewer 2 Report

Congratulations on your work. It would be nice to explain why did you choose particular value of LVEF in your research, maybe you should cite the ESC guidelines. 

Author Response

Reply: We thank the reviewer for this comment and added a relevant reference as per this request. The paragraph (line 82) now read:
“This cut-off was based on the European Society of Cardiology definition for preserved left-ventricular EF”

Reviewer 3 Report

This is an interesting manuscript involving the burgeoning field of attempting to use “neural networks” (aka “Artificial intelligence”) for enhanced, 12-lead ECG-based detection of anatomical heart conditions, in this case of left ventricular systolic dysfunction. The manuscript is “generally” well-written, with its greatest strength in my view being its incorporation of the manual determinations by cardiologists for comparison purposes.

However, some critical comments and suggestions are as follows:

First, due to the common occurrence of diastolic heart failure, aka heart failure with preserved ejection fraction or “HFpEF”, which is also principally a left ventricular pathology, I’d suggest that the authors change their “LVD” abbreviation to “LVSD” (left ventricular SYSTOLIC dysfunction) instead, meaning wherever that former abbreviation currently occurs, to make it clearer to readers exactly what the authors have studied and what they have not studied.

Second, it’s highly disputable, especially today, that “The gold standard for diagnosis of asymptomatic LV[S]D is echocardiography” (e.g., lines 38 and 268 of the manuscript). Most astute clinicians today would more correctly assert that Cardiac MRI (CMR) is THE diagnostic gold standard for any anatomical type of LVSD, whether symptomatic or not, whereas echocardiography can still be “A” gold standard for scientific purposes if one has nothing else, as was clearly the case in this study. Thus I’d suggest that the authors delete all current statements suggesting that echocardiography is “THE” gold standard for detecting reduced LVEF (e.g., lines 38 and 268 of the manuscript), while replacing such statements, if/when needed, with statements to the effect that “we [the authors] used echocardiography as OUR [my emphasis] gold standard”, etc.

Third, the authors have not noted, within their Results section, how many ECG files in their training, validation and test sets, respectively, had complete bundle branch blocks (both left and right). Nor have they noted whether any methodological measures were put into place to attempt to equalize the percentages of such BBBs within their low EF versus normal EF groups. Nor have they noted the extent to which the presence of BBBs, or of other baseline ECG abnormalities that they HAVE reported, e.g., atrial fibrillation, might have affected the results both of their neural network’s outputs, and of their physicians’ manual readings, especially if there did end up being large differentials in the numbers/percentages of BBBs between groups (as there are in AF-related percentages). For example, to what extent might just a theoretically higher prevalence of LBBB in one group or the other have potentially biased their CNN to detect (or to not detect) BBB, rather than LVSD per se? Readers will never be able know the answer to that question, nor the extent to which the authors’ CNN is generalisable to other data sets that have some other percentage of BBBs or perhaps disallow BBBs altogether, unless readers are specifically informed. Thus the authors must also add some related results, as well as some further discussion, addressing these important issues.

Fourth, the authors have not recognized or cited any pre-existing literature wherein LVSD has been possibly more accurately detected through the use “discrete measures of conventional and advanced 12-lead ECG”, meaning with assistance from ML and from more traditionally robust statistics, but wherein no form of “[deep or other] neural network”, or any form of black box “artificial” intelligence per se, has been required. This is important because there are also a number of scientific and ethical limitations involving the use of “neural network-based artificial intelligence” in the clinical setting that the authors have failed to recognize and discuss in their manuscript, but that they must discuss, ideally within a Limitations section or equivalent.

Included first below are links to three publications wherein LVSD has been accurately detected through the use of discrete-measure advanced 12-lead ECG. The last of these three is recent, but with the oldest of the three dating back to more than a decade ago, meaning long before the scientifically and ethically more problematic, (and also "less rigorous", because they fail to even attempt the more rigorous digital signal processing required) “artificial intelligence” techniques became de rigueur. Note that the middle of the three publications below also contains a comparison to the performance of physician-read 12-lead ECGs, similar to what the authors have done in their own study. In any case, the authors should cite and discuss at least 1-2 of these prior publications, discussing the their prior results in light of their own results:

https://www.ncbi.nlm.nih.gov/pmc/articles/PMC2894002/

https://www.mdpi.com/2308-3425/2/2/93/htm

https://www.futuremedicine.com/doi/full/10.2217/fca-2020-0225

And in relation to the above, the authors should also include some discussion, presumably within their Limitations section, about why a neural network-based technique like theirs might be at a disadvantage to the more classical techniques involved in the publications above. That is, not only at a disadvantage in relation to “interpretability”, but also in relation to “ethical accountability”, both being compromised whenever “black box”, neural network-type AI techniques are employed, due to their inherent lack of transparency and explainability. The authors should also note that it’s effectively impossible for a clinician to identify, when critically evaluating the diagnostic output of any neural network-based AI model, the contribution to the result from methodological artifact or bias merely related to noise or to differing technical specifications between different ECG machines. But whether or not authors are already intimately familiar with these issues, I’d strongly suggest that they carefully read the all of additional publications below, while also citing and discussing at least a couple of them (in relation to their potential implications for their own current work) within their Limitations section:   

The Lancet Respiratory Medicine. Opening the black box of machine learning. Lancet Resp Med 6, 801–801, doi:10.1016/S2213-2600(18)30425-9 (2018).

Yoon, C. H., Torrance, R. & Scheinerman, N. Machine learning in medicine: should the pursuit of enhanced interpretability be abandoned? J Med Ethics, doi:10.1136/medethics-2020-107102 (2021).

Brisk, R. et al. The effect of confounding data features on a deep learning algorithm to predict complete coronary occlusion in a retrospective observational setting. Eur Heart J - Dig Health 2, 127–134, doi:10.1093/ehjdh/ztab002 (2021).

Siontis, K. C. et al. Use of Artificial Intelligence Tools Across Different Clinical Settings: A Cautionary Tale. Circulation. Cardiovasc Qual Outcomes 14, e008153, doi:10.1161/circoutcomes.121.008153 (2021).

Lindow, T. et al. Heart age estimated using explainable advanced electrocardiography. Sci Rep2022 Jun 14;12(1):9840. doi: 10.1038/s41598-022-13912-9. [See especially this publication’s Discussion section]. 

Volovici, V. et al. Steps to avoid overuse and misuse of machine learning in clinical research. Nat Med (2022). https://doi.org/10.1038/s41591-022-01961-6

Author Response

This is an interesting manuscript involving the burgeoning field of attempting to use “neural networks” (aka “Artificial intelligence”) for enhanced, 12-lead ECG-based detection of anatomical heart conditions, in this case of left ventricular systolic dysfunction. The manuscript is “generally” well-written, with its greatest strength in my view being its incorporation of the manual determinations by cardiologists for comparison purposes.

However, some critical comments and suggestions are as follows:

First, due to the common occurrence of diastolic heart failure, aka heart failure with preserved ejection fraction or “HFpEF”, which is also principally a left ventricular pathology, I’d suggest that the authors change their “LVD” abbreviation to “LVSD” (left ventricular SYSTOLIC dysfunction) instead, meaning wherever that former abbreviation currently occurs, to make it clearer to readers exactly what the authors have studied and what they have not studied.

Reply: We wish to thank the reviewer for his/her suggestions. As for this specific comment, LVD was replaced with LVSD throughout the manuscript.

Second, it’s highly disputable, especially today, that “The gold standard for diagnosis of asymptomatic LV[S]D is echocardiography” (e.g., lines 38 and 268 of the manuscript). Most astute clinicians today would more correctly assert that Cardiac MRI (CMR) is THE diagnostic gold standard for any anatomical type of LVSD, whether symptomatic or not, whereas echocardiography can still be “A” gold standard for scientific purposes if one has nothing else, as was clearly the case in this study. Thus I’d suggest that the authors delete all current statements suggesting that echocardiography is “THE” gold standard for detecting reduced LVEF (e.g., lines 38 and 268 of the manuscript), while replacing such statements, if/when needed, with statements to the effect that “we [the authors] used echocardiography as OUR [my emphasis] gold standard”, etc.

Reply: We agree with the reviewer and the relevant paragraphs now read:
Echocardiography, a non-invasive, ultrasound-based exam, is the recommended key investigation for the assessment of cardiac function.” Line 38 references and cites the 2021 ESC guidelines for heart failure. and “Taken together, since echocardiography is still the recommended key investigation exam for the diagnosis of cardiac dysfunction and this test is not readily available” in line 268

Third, the authors have not noted, within their Results section, how many ECG files in their training, validation and test sets, respectively, had complete bundle branch blocks (both left and right). Nor have they noted whether any methodological measures were put into place to attempt to equalize the percentages of such BBBs within their low EF versus normal EF groups. Nor have they noted the extent to which the presence of BBBs, or of other baseline ECG abnormalities that they HAVE reported, e.g., atrial fibrillation, might have affected the results both of their neural network’s outputs, and of their physicians’ manual readings, especially if there did end up being large differentials in the numbers/percentages of BBBs between groups (as there are in AF-related percentages). For example, to what extent might just a theoretically higher prevalence of LBBB in one group or the other have potentially biased their CNN to detect (or to not detect) BBB, rather than LVSD per se? Readers will never be able know the answer to that question, nor the extent to which the authors’ CNN is generalisable to other data sets that have some other percentage of BBBs or perhaps disallow BBBs altogether, unless readers are specifically informed. Thus the authors must also add some related results, as well as some further discussion, addressing these important issues.

Reply: We thank the reviewer for this important and relevant comment. As opposed to physicians, who classify the ECGs by patterns, the classification made by DNN is based on a collection of pixels and not patterns. Thus, specific ECG pathologies (e.g., Q waves, bundle branch blocks, etc.) are not part of the non-structured learning process leading to classification per se. A similar methodology was utilized by Attia et. al in their highly cited manuscript published in Nature Medicine (citation #17 in the references list). Since our dataset included over 200,000 ECGs without any automated or human interpretation/labeling we weren’t able to detail in the baseline characteristics table pathologies mentioned by the reviewer and to explore the impact of these pathologies on the learning process. It should be noted that in a cohort of this magnitude, the odds that discrepancies in the prevalence of specific ECG pathologies are probably low. On the other hand, we’ve attempted to explore the patterns leading to the physicians’ decision to classify an ECG as “normal” or “abnormal” EF but the small sample size and the heterogeneity between physicians hampered the ability to draw meaningful conclusions. We are currently engaged in a study aimed at exploring the impact of certain ECG patterns on the process of classification, but this requires a significant amount of time and resources, and we thus opted to address this in a separate report. We acknowledged this relevant limitation in the “limitations” section as follows:” First, the parameters used by MLA for classification of ECGs in this study are unknown. This inherent “black box” feature of MLA prevented us from highlighting specific traditional (e.g., bundle branch blocks, Q waves, etc.) and non-traditional, unknown parameters associated with LVSD. Beyond the inability to gain new insights regarding unknown features associated with LSVD, this lack of identifiable parameters further limits our ability to retrain physicians and improve their performance. Although less likely in a large cohort, it is also possible that some of the determinants for classification by the MLA are based on artifacts or features relevant only to the machines used in this study. These dataleaks and other limitations concerning the use of MLA were discussed and highlighted by prior reports in this field of research"

Fourth, the authors have not recognized or cited any pre-existing literature wherein LVSD has been possibly more accurately detected through the use of “discrete measures of conventional and advanced 12-lead ECG”, meaning with assistance from ML and from more traditionally robust statistics, but wherein no form of “[deep or other] neural network”, or any form of black box “artificial” intelligence per se, has been required. This is important because there are also a number of scientific and ethical limitations involving the use of “neural network-based artificial intelligence” in the clinical setting that the authors have failed to recognize and discuss in their manuscript, but that they must discuss, ideally within a Limitations section or equivalent.

Included first below are links to three publications wherein LVSD has been accurately detected through the use of discrete-measure advanced 12-lead ECG. The last of these three is recent, but with the oldest of the three dating back to more than a decade ago, meaning long before the scientifically and ethically more problematic, (and also "less rigorous", because they fail to even attempt the more rigorous digital signal processing required) “artificial intelligence” techniques became de rigueur. Note that the middle of the three publications below also contains a comparison to the performance of physician-read 12-lead ECGs, similar to what the authors have done in their own study. In any case, the authors should cite and discuss at least 1-2 of these prior publications, discussing the their prior results in light of their own results:

https://www.ncbi.nlm.nih.gov/pmc/articles/PMC2894002/

https://www.mdpi.com/2308-3425/2/2/93/htm

https://www.futuremedicine.com/doi/full/10.2217/fca-2020-0225

Reply: We thank the reviewer for pointing out these studies. We’ve decided to include them in the discussion section (line 241) which reads:
“More recently, “advanced ECG” (aECG) was demonstrated to have high specificity and sensitivity for LVSD diagnosis. This technology utilizes computerized analysis of the standard 12-lead ECG that incorporates spatial and temporal data that exceeds the data provided by the standard ECG analysis software.[31,32] Of note, a relatively small-scale study demonstrated that aECG outperformed physicians in predicting LVSD in a cohort of 79 patients.[33]

And in relation to the above, the authors should also include some discussion, presumably within their Limitations section, about why a neural network-based technique like theirs might be at a disadvantage to the more classical techniques involved in the publications above. That is, not only at a disadvantage in relation to “interpretability”, but also in relation to “ethical accountability”, both being compromised whenever “black box”, neural network-type AI techniques are employed, due to their inherent lack of transparency and explainability. The authors should also note that it’s effectively impossible for a clinician to identify, when critically evaluating the diagnostic output of any neural network-based AI model, the contribution to the result from methodological artifact or bias merely related to noise or to differing technical specifications between different ECG machines. But whether or not authors are already intimately familiar with these issues, I’d strongly suggest that they carefully read the all of additional publications below, while also citing and discussing at least a couple of them (in relation to their potential implications for their own current work) within their Limitations section:   

The Lancet Respiratory Medicine. Opening the black box of machine learning. Lancet Resp Med 6, 801–801, doi:10.1016/S2213-2600(18)30425-9 (2018).

Yoon, C. H., Torrance, R. & Scheinerman, N. Machine learning in medicine: should the pursuit of enhanced interpretability be abandoned? J Med Ethics, doi:10.1136/medethics-2020-107102 (2021).

Brisk, R. et al. The effect of confounding data features on a deep learning algorithm to predict complete coronary occlusion in a retrospective observational setting. Eur Heart J - Dig Health 2, 127–134, doi:10.1093/ehjdh/ztab002 (2021).

Siontis, K. C. et al. Use of Artificial Intelligence Tools Across Different Clinical Settings: A Cautionary Tale. Circulation. Cardiovasc Qual Outcomes 14, e008153, doi:10.1161/circoutcomes.121.008153 (2021).

Lindow, T. et al. Heart age estimated using explainable advanced electrocardiography. Sci Rep. 2022 Jun 14;12(1):9840. doi: 10.1038/s41598-022-13912-9. [See especially this publication’s Discussion section]. 

Volovici, V. et al. Steps to avoid overuse and misuse of machine learning in clinical research. Nat Med (2022). https://doi.org/10.1038/s41591-022-01961-6

Reply: We thank again for the reviewer for this thorough work in attempting to assist us in delivering the main messages from our study while highlighting the limitations of this analysis. We carefully read all the above citations. As presented below, we’ve included a discussion regarding the inherent limitation of using MLA in the first limitation mentioned and three of the manuscripts were cited: “First, the parameters used by MLA for classification of ECGs in this study are unknown. This inherent “black box” feature of MLA prevented us from highlighting specific traditional (e.g., bundle branch blocks, Q waves, etc.) and non-traditional, un-known parameters associated with LVSD. Beyond the inability to gain new insights regarding unknown features associated with LSVD, this lack of identifiable parameters further limits our ability to re-train physicians and improve their performance. Although less likely in a large cohort, it is also possible that some of the determinants for classification by the MLA are based on artifacts or features relevant only to the machines used in this study. These dataleaks and other limitations concerning the use of MLA were discussed and highlighted by prior reports in this field of research.”

Round 2

Reviewer 3 Report

The authors have done a nice job with their responses and revision. And especially with their new additions and limitations as noted, the manuscript is now acceptable to me.